

# Prenatal diagnosis of fetuses conceived by assisted reproductive technology by karyotyping and chromosomal microarray analysis

Huan Guo[1], Rui Sheng[1], Xiu Zhang[2], Xuemei Jin[2], Wenjing Gu[1], Ting Liu[1], Haixin Dong[1] and Ran Jia[2]

[1] Department of Clinical Laboratory, Affiliated Hospital of Jining Medical University, Jining, Shandong, China
[2] Department of Obstetrics, Affiliated Hospital of Jining Medical University, Jining, Shandong, China

## ABSTRACT

**Background**. Invasive prenatal evaluation by chromosomal microarray analysis (CMA) and karyotyping might represent an important option in pregnant women, but limited reports have applied CMA and karyotyping of fetuses conceived by assisted reproductive technology (ART). This study aimed to examine the value of CMA and karyotyping in prenatal diagnosis after ART.

**Methods**. This retrospective study included all singleton fetuses conceived by ART from January 2015 to December 2021. Anomalies prenatally diagnosed based on karyotyping and CMA were analyzed. Prevalence rates for various CMA and karyotyping results were stratified based on specific testing indications including isolated—and non-isolated ART groups. The rates of CMA findings with clinical significance (pathogenic/likely pathogenic) and karyotype anomalies were assessed and compared to those of local control individuals with naturally conceived pregnancies and without medical indications.

**Results**. In total, 224 subjects were assessed by karyotyping and CMA. In the examined patients, chromosomal and karyotype abnormality rates were 3.57% (8/224) and 8.93% (20/224), respectively. This finding indicated a 5.35% (12/224)-incremental rate of abnormal CMA was obtained over karyotype analysis ($p = 0.019$). The risk of CMA with pathogenic findings for all pregnancies conceived by ART (5.80%, 13/224) was markedly elevated in comparison with the background value obtained in control individuals (1.47%, 9/612; $p = 0.001$). In addition, risk of CMA with clinically pathogenic results in isolated ART groups was significant higher compared to the background risk reported in the control cohort ($p = 0.037$).

**Conclusions**. Prenatal diagnosis including karyotyping and CMA is recommended for fetuses conceived by ART, with or without ultrasound findings.

Corresponding author
Ran Jia, huankwo@163.com

## INTRODUCTION

Because of recent advances in assisted reproductive technology (ART), a rising number of pregnancies result from related procedures. Experts estimate that there will be 167 million individuals conceived by ART by 2100 (*Faddy, Gosden & Gosden, 2018*).

ART has always been debatable. Despite its demonstrated safety, follow-up analysis of individuals conceived by ART indicated the related procedures may promote epigenetic, genetic and/or developmental anomalies (*Belva et al., 2020*; *Cui et al., 2021*; *Hattori et al., 2019*; *Long et al., 2020*). Multiple studies across mammalian species have established that *in vitro*-derived embryos have remarkably frequent subchromosomal losses/gains and chromosome instability (*Daughtry et al., 2019*; *Tsuiko et al., 2017*). However, evidence is still lacking about microscopic/submicroscopic copy number variations for fetuses conceived by assisted reproductive technology. Chromosomal microarray analysis (CMA) is considered a first-tier method for detecting structural anomalies of the fetus, identifying microscopic/submicroscopic copy number variations (CNVs). Currently, non-invasive prenatal testing (NIPT) has high popularity among ART-treated women because of no procedure-associated risk of miscarriage unlike chorion villus sampling and amniocentesis (maximum of 0.5%) (*Salomon et al., 2019*; *Wulff et al., 2016*). However, NIPT is considered an important technique for detecting fetal aneuploidies, with elevated detection rates for trisomy 21 (99%), trisomy 18 (96%) and trisomy 13 (91%), but does not detect other chromosomal abnormalities and microscopic/submicroscopic copy number variations (*Taylor-Phillips et al., 2016*). G-banding karyotype analysis has been predominantly utilized for detecting chromosomal abnormalities clinically in recent decades. However, this technique has low resolution and is time-consuming. Currently, only few reports have applied CMA and G-banding karyotyping in fetuses conceived by ART. The current work aimed to assess whether ART increases CNV and karyotype abnormality rates, determining the value of CMA and G-banding karyotyping in prenatal analysis of fetuses conceived by assisted reproductive technology in comparison with naturally-conceived fetuses. Our hypothesis was that ART increases CNV and karyotype abnormality rates, and that CMA analysis will show a higher incidence of abnormal findings compared with previous reports using microscopic chromosomal testing.

## MATERIALS & METHODS

### Study design and participants

A retrospective cross-sectional study was carried out to assess fetuses conceived by assisted reproductive technologies delivered between January 2015 and December 2021 at the Affiliated Hospital of Jining Medical University, a tertiary hospital, in Shandong, China. The study population included singleton pregnant women who conceived *via* ART procedures with invasive genetic testing by both karyotyping and CMA. All genetic samples were obtained by amniocentesis. Inclusion criteria were pregnancy conceived *via* ART procedures in women, with or without other abnormal medical indications, single fetus and invasive genetic tests (both karyotyping and CMA). Exclusion criteria were natural

conception in women, multiple pregnancy, no first-trimester ultrasound, no invasive genetic tests (both karyotyping and CMA) only karyotype analysis or CMA.

Based on ultrasound findings and other medical indications, the enrolled pregnant women were classified into two groups: ART only (isolated ART group) and ART accompanied by soft ultrasound markers, ultrasound malformations and other medical indications (non-isolated ART group). In the non-isolated ART group, the most common medical indications were abnormal ultrasound findings (AUS), including structural abnormalities, increased nuchal translucency, intrauterine growth restriction, intestinal hyperechogenicity and amniotic fluid abnormalities, etc, followed by elevated odds of maternal serum Down syndrome screening, confirmation of a known anomalous fetal NIPT, a family history of a genetic disease or chromosomal alteration and adverse pregnancy history including trisomy 21 reproductive history and a reproductive history of a genetic condition or chromosomal abnormality.The control population was a group of individuals with naturally conceived fetuses and no medical indications. Pregnancies without medical indications underwent CMA and karyotyping by maternal request and elevated maternal age in Affiliated Hospital of Jining Medical University. There were no other indications for performing invasive diagnostics at the subgroup that underwent solely ART and at the control group.

## Follow-up of pregnancy outcomes

Clinical follow-up assessments of pregnancy outcome, prenatal and postnatal development were performed regularly by telephone.

## Karyotype analysis

Ultrasound-guided amniocentesis was carried out at pregnancy weeks 17–28, collecting amniotic fluid samples (30 ml each). Then, 20 ml amniotic fluid samples were assessed based on the amniotic fluid karyotyping procedure of the prenatal diagnosis department of our hospital (320 to 400 bands). All prenatal samples were routinely cultured, mounted on slides and subjected to G-banding (additional C-banding and N-banding if required). Karyotype analysis was performed on a GSL-120 Streamlines Cytogenetic Analysis System (Leica Microsystems; Mannheim, Germany). At least 40 karyotypes were counted for each case, and five karyotypes were randomly selected for analysis.

## CMA analysis

The remaining 10 ml amniotic fluid is used for chromosomal microarray analysis. DNA extraction was performed with the QIAamp DNA Blood Kit (QIAGEN, Hilden, Germany) following the kit's handbook (http://www.qiagen.com). CMA was carried out by high-resolution genotyping single nucleotide polymorphism microarray with Affymetrix CytoScan 750k Array (Affymetrix, USA). CNV analysis was based on findings reported for the human reference genome 37 (NCBI37hg19) by the National Centre for Biotechnology Information. CNV was assessed by reviewing multiple databases, including DGV (http://dgv.tcag.ca/dgv), OMIM (https://omim.org/), DECIPHER (https://decipher.sanger.ac.uk/), PubMed (https://www.ncbi.nlm.nih.gov/pubmed/) and others.

Next, CMA findings were categorized as follows: (1) clinically significant, *i.e.,* pathogenic/likely pathogenic; (2) variants of unknown significance (VOUS) (findings of unknown significance or CNV with clinical penetrance below 10%, *e.g.*, duplications at 15q13.3, 16p11.2 and 16p11.13 loci) (*Maya et al., 2018*; *Rosenfeld et al., 2013*); (3) normal findings, *i.e.,* no CNV, benign/likely benign CNVs, or VOUS findings below the reported cutoffs of 1 and 2 Mb for deletions and duplications, respectively.

## Statistical analysis

SPSS 20.0 (IBM, Armonk, NY, USA) was utilized for data analysis. Continuous and categorical data were expressed as mean ± standard deviation and frequency or percentage, respectively; they were compared by the Student's t test and the chi-square test or Fisher's exact test, respectively. $P < 0.05$ indicated statistical significance.

## Ethics approval and consent to participate

This study was approved by the Ethics Committee of the Affiliated Hospital of Jining Medical University, and the ethics approval number is 2020c052. The date of approval was 2 December 2020. All subjects signed an informed consent form.

# RESULTS

The study included 224 fetuses conceived by ART between January 1, 2015 and December 31, 2021. The pregnant women averaged 34 years old (range, 20 to 51 years), and the mean gestational age was 20 weeks (range, 17 to 28 weeks). Of the 224 fetuses, there were 84 and 140 in the isolated- and non-isolated ART groups, respectively. In the isolated ART group, fetuses conceived by ART only, without other abnormalities. In the non-isolated ART group, fetuses conceived by ART had other abnormalities, including soft ultrasound markers, ultrasound malformations and adverse pregnancy history. In 612 control individuals, CMA testing yielded nine pathogenic CNVs (1.47%) and karyotype testing yielded three aneuploidy cases (0.49%). The maternal characteristics and abnormal results of the study group are shown in Table 1 and Fig. 1.

## Prenatal diagnostic karyotyping data

Karyotyping in all ART groups revealed chromosomal anomalies in eight fetuses, with seven aneuploidy cases and one unbalanced translocation. The clinico-genetic features of fetuses with abnormal karyotypes are shown in Table 2. Precisely, chromosomal alterations were found in 3.57% of individuals (8/224). The commonest chromosomal alterations included trisomy 21 (1.79%, 4/224) and sex chromosome abnormalities (1.79%, 4/224). Sex chromosome abnormalities included Klinefelter's syndrome, hyperestrogenism, hyperandrogenism and unbalanced sex chromosome translocation. Compared with the control population, there was elevated rate of chromosomal anomalies in the total ART group (0.49 *vs.* 3.57%, $\chi^2 = 11.990$; $p = 0.001$, Chi-square test) (Table 1).

In the isolated ART group, there was no chromosomal abnormality. In the non-isolated ART group, chromosomal anomalies were found in 8 fetuses. In comparison with the isolated ART group, the non-isolated ART group had starkly elevated rate of chromosomal

**Table 1 Maternal characteristics, abnormal karyotyping and abnormal CMA findings in the study population.**

| | ART | | | Control |
| | All ART (N = 224) | Non-isolated ART (N = 140) | Isolated ART (N = 84) | N = 612 |
| --- | --- | --- | --- | --- |
| Maternal age (years) | 33.75 ± 5.49 (20–51) | 32.85 ± 5.47 (21–51) | 35.10 ± 4.92 (20–44) | 37.87 ± 3.94 (20–48) |
| Gestation age at invasive testing (weeks) | 19.64 ± 1.64 (17-28) | 19.64 ± 1.70 (17–28) | 19.70 ± 1.58 (18–28) | 19.45 ± 1.40 (17–27) |
| Aneuploidies | 8/224(3.57)[a] | 8/140 (5.71)[a,b] | 0/84(0) | 3/612 (0.49) |
| T21 | 4/224 (1.79) | 4/140 (2.86) | 0/84 (0) | 1/612 (0.16) |
| Sex chromosome abnormalities | 4/224 (1.79) | 4/140 (2.86) | 0/84(0) | 2/612 (0.33) |
| CMA | 20/224(8.93) | 14/140 (10) | 6/84(7.14)[c] | 41/612(6.70) |
| Pathogenic CNVs | 13/224(5.80)[a] | 9/140 (6.43)[a] | 4/84(4.76)[a,c] | 9/612 (1.47) |
| VOUS | 7/224(3.13) | 5/140 (3.57) | 2/84(2.38) | 32/612 (5.23) |

Notes.

ART, assisted reproductive technology; CMA, chromosomal microarray analysis; VOUS, Variants of unknown significance.

[a]Significant difference versus CONTROL; $P < 0.05$.

[b]Significant difference versus isolated ART, $P < 0.05$.

[c]Significant difference versus Aneuploidies, $P < 0.05$.

**Table 2 Type of abnormalities detected and clinical relevant characteristics in fetuses with an abnormal karyotype.**

| NO | Other anomalies | Karyotyping results | CMA | Pregnancy outcome |
| --- | --- | --- | --- | --- |
| 1 | Nuchal translucency thickening | 47,XX, +21 | arr(21) × 3 | TOP |
| 2 | Advanced maternal age, cleft lip and palate | 47,XY, +21 | arr(21) × 3 | TOP |
| 3 | T21 high risk of maternal serum Down syndrome screening, Echogenic bowel, intracardiac echogenic foci | 47,XX, +21 | arr(21) × 3 | TOP |
| 4 | Advanced maternal age, Echogenic bowel, intracardiac echogenic foci | 47,XY, +21 | arr(21) × 3 | TOP |
| 5 | Vanishing twin | 47,XYY | arr(X)x1,(Y)X2 | Born |
| 6 | Nuchal translucency thickening | 47,XXX | arr(X) × 3 | TOP |
| 7 | Nuchal translucency thickening | 46,XX,i(X)(q10) | arr[GRCh37]Xp22.33p11.21 (168551_57884399) × 1, Xp11.21q28 (57888525_155233098) × 3 | TOP |
| 8 | Advanced maternal age, vanishing twin | 47,XXY | arr(X) × 2,(Y) × 1 | TOP |

Notes.

CMA, chromosomal microarray analysis; TOP, Termination of pregnancy.

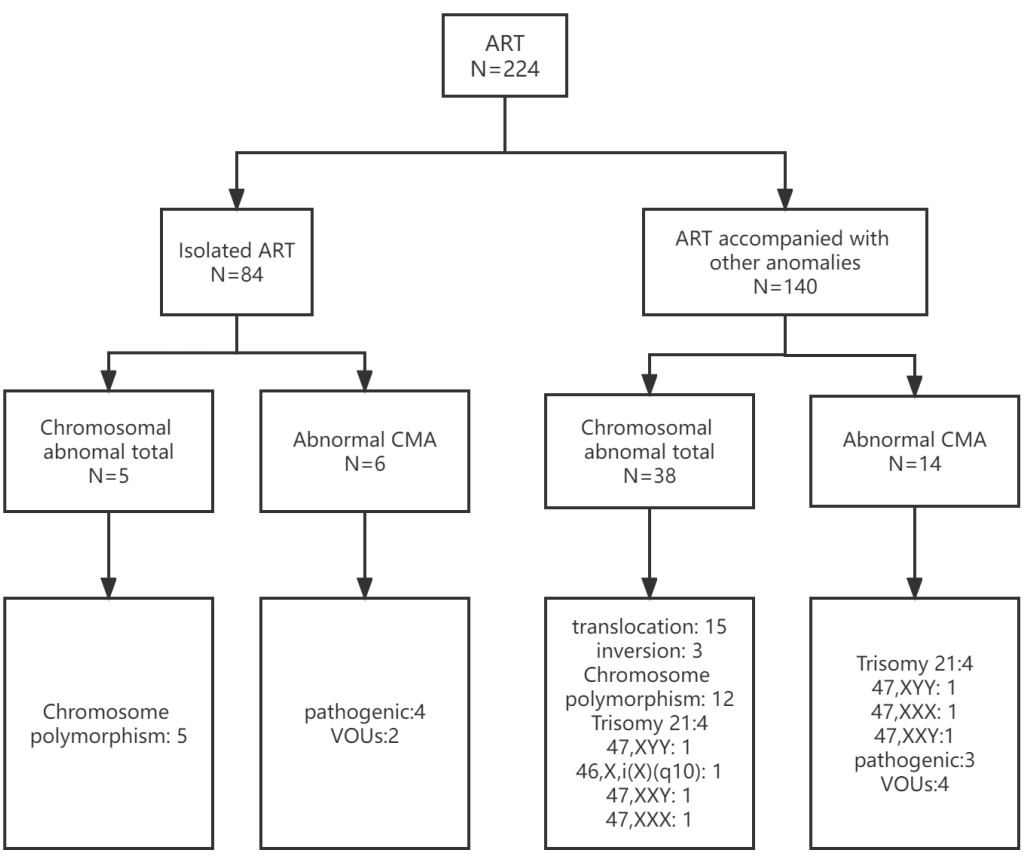

**Figure 1  Prenatal diagnosis in the isolated and non-isolated ART groups.** ART, assisted reproductive technology; CMA, chromosomal microarray analysis; VOUS, variants of unknown significance.

anomalies (0 *vs.* 5.71%, $\chi^2 = 4.978$; $p = 0.026$, Chi-square test). In comparison with the control population, chromosomal abnormalities had no starkly elevated rate in the isolated ART group (0.49 *vs.* 0%, $\chi^2 = 0.414$; $p = 0.520$, Chi-square test). In comparison with the control population, there was markedly elevated rate of chromosomal alterations in the non-isolated ART group (0.49 *vs.* 5.71%, $\chi^2 = 21.573$; $p = 0.000$, Chi-square test) (Table 1).

## Prenatal diagnostic findings by CMA

Pathogenic/likely pathogenic CNVs were detected in 13/224 fetuses, indicating a total detection rate in CMA for genetic alterations of 5.80%, including seven aneuploidies and one unbalanced sex translocation. Of the 216 fetuses with normal karyotype, five showed pathogenic/likely pathogenic CNVs. Totally seven fetuses were detected with VOUS (Table 3).

In case 1, a nulliparous 31-year-old woman with a low-risk pregnancy showed unremarkable first- and second-trimester ultrasound findings. The karyotype was normal, but CMA revealed a pathogenic CNV, showing a duplication in 16q24.1q24.3 that may cause intellectual disability, generalized hypotonia and global developmental delay. In

**Table 3  Type of abnormalities detected and clinical relevant characteristics in fetuses with normal karyotyping and chromosomal microarray analysis findings.**

| NO | Other anomalies | CMA | Size | CNV type | Categorization | Pregnant outcome |
|---|---|---|---|---|---|---|
| 1 | Parental request | arr[GRCh37] 16q24.1q24.3(85958373-90119719) × 3 | 4.16Mb | Gain | Pathogenetic | Live birth |
| 2 | Advanced maternal age | arr[GRCh37] 17p12(14073535_15482833) × 1 | 1409kb | Loss | Pathogenetic | Live birth |
| 3 | Parental request | arr[GRCh37] 1q21.1q21.2(146106723_148016122) × 1 | 1909kb | Loss | Pathogenetic | Live birth |
| 4 | Advanced maternal age | arr[GRCh37] 17q12(34440088_36243365) × 3 | 1803kb | Gain | Pathogenetic | TOP |
| 5 | High risk of fetal T18 after Down syndrome screening | arr[GRCh37] Xp22.31(6679109_8125388) × 0 | 1446kb | Loss | Pathogenetic | Live birth |
| 6 | Balanced chromosomal structural abnormalities from mother | arr[GRCh37] 4q35.2(188936538-189769264) × 1 | 832.73kb | Loss | VOUS | Live birth |
| 7 | Chromosomal inversion | arr[GRCh37] 15q11.2(22822019-23085218) × 1 | 263.2kb | Loss | VOUS | Live birth |
| 8 | Advanced maternal age | arr[GRCh37] 6p25.1(4216798-5606700) × 3 | 1.39Mb | Gain | VOUS | Live birth |
| 9 | Choroid plexus cysts | arr[GRCh37] 1p31.1(72543979-74089027) × 3 | 1.55Mb | Gain | VOUs | Live birth |
| 10 | Ghromosomal inversion advanced maternal age | arr[GRCh37] 2p13.2p12(73019188-75209688) × 3 | 2.19Mb | Gain | VOUs | Live birth |
| 11 | Advanced maternal age | arr[GRCh37] 7p15.3p15.2(22720487_25743493) × 1,12q24.33(133156066_133663089) × 1 | 3.02Mb, 507kb | Loss | VOUS | Live birth |
| 12 | Balanced chromosomal structural abnormalities from father, Choroid plexus cysts, T21 critical risk of Down syndrome screening | arr[GRCh37] 16p13.13p13.12(12056151_14420065) × 1 | 2364kb | Loss | VOUS | Live birth |

**Notes.**
CMA, chromosomal microarray analysis; VOUS, Variants of unknown significance; TOP, Termination of pregnancy.

case 2, CMA showed a microdeletion in 17p12, which may be involved in hereditary motor neuropathy with liability to pressure palsies (HNPP). In case 3, the 1q21.1q21.2 recurrent microdeletion was found and could lead to microcephaly, moderate cognitive impairment, moderate dysmorphic facial traits, eye abnormalities and cardiac defect. In case 4, CMA showed a duplication in 17q12 that may cause microcephaly, short stature, developmental delays, and renal and cardiac abnormalities. In case 5 (a male fetus with high risk of trisomy 18), CMA revealed a duplication in Xp22.31 that might induce ichthyosis, intellectual disability and seizure.

In the total ART group, CMA anomalies were detected in 20 fetuses, including 13 pathogenic/likely pathogenic CNVs and seven VOUS. In the isolated ART group, CMA anomalies were detected in six fetuses, including four pathogenic/likely pathogenic CNVs and two VOUS. In the non-isolated ART group, CMA anomalies were detected in 14 fetuses, including nine pathogenic/likely pathogenic CNVs and five VOUS. Compared with the control population, the total ART group had no starkly elevated rate of CMA anomalies (6.70 *vs.* 8.93%, $\chi^2 = 1.205$; $p = 0.272$, Chi-square test). Compared with control individuals, no starkly elevated rate of CMA anomalies was detected in the isolated ART group (6.70 *vs.* 7.14%, $\chi^2 = 0.023$; $p = 0.879$, Chi-square test). Compared with control individuals, CMA anomalies had remarkably higher rate in the non-isolated ART group (6.70 *vs.* 10%, $\chi^2 = 1.831$; $p = 0.176$, Chi-square test). However, compared with the control group, the prevalence of pathogenic/likely pathogenic CNVs showed a significant increase in the isolated ART group (1.47 *vs.* 4.76%, $\chi^2 = 4.365$; $p = 0.037$, Chi-square test). In comparison with control individuals, the non-isolated ART group had markedly elevated rate of pathogenic/likely pathogenic CNVs (1.47 *vs.* 6.43%, $\chi^2 = 11.988$; $p = 0.001$, Chi-square test). In comparison with the isolated ART group, pathogenic/likely pathogenic CNVs had non-significantly elevated rate in the non-isolated ART group (4.76 *vs.* 6.43%, $\chi^2 = 0.267$; $p = 0.606$, Chi-square test). The prevalence rates of VOUS were similar among groups ($P > 0.05$). Using pairwise comparison methods to assess VOUS, nonsignificant differences were found between the total ART and control groups and among isolated ART, non-isolated ART, and control groups ($p > 0.05$) (Table 1).

Comparing the detection rate of abnormalities by karyotyping, the detection rate of genetic abnormalities detected by CMA was increased in the non-isolated ART group, but with a nonsignificant difference (5.71 vs.10%, $\chi^2 = 1.776$; $p = 0.183$, Chi-square test); however, the rate of genetic alterations detected by CMA was significantly increased in the isolated ART group (0 *vs.* 7.14%, $\chi^2 = 6.222$; $p = 0.013$, Chi-square test) .CMA markedly elevated the diagnostic yield of pathogenic/likely pathogenic anomalies in fetuses conceived by assisted reproductive technologies compared to karyotyping(0 *vs.* 4.76%, $\chi^2 = 4.098$; $p = 0.043$, Chi-square test) (Table 1).

## DISCUSSION

Compared with control individuals, chromosomal anomalies showed a nonsignificant difference in the isolated ART group (0.49 *vs.* 0%, $\chi^2 = 0.414$, $p = 0.520$) but a significant increase in the non-isolated ART group (0.49 *vs.* 5.71%, $\chi^2 = 21.573$, $p = 0.000$) in this study. In the prenatal cohort, karyotyping revealed an elevated rate of chromosomal abnormalities in the fetuses of the non-isolated ART group compared with the isolated ART group (0 *vs.* 5.71%, $\chi^2 = 4.978$; $p = 0.026$), which is likely because multiple ultrasound anomalies, not ART itself, are soft markers of aneuploidy, *e.g.*, increased NT, echogenic bowel, intracardiac echogenic foci and cleft lip and palate. In addition, in this study, two aneuploidy cases were found in vanishing twin syndrome. Further large trials assessing vanishing twin syndrome are required to confirm soft markers for aneuploidy. In the isolated ART group, relatively few patients were examined. With this small sample size,

there were very few instances of abnormal findings in pregnancies. Therefore, more samples should be used to explore if ART causes causes chromosomal anomalies.

Compared with control individuals, the prevalence of pathogenic CNVs in the isolated ART group showed a significant increase (1.47 *vs.* 4.76%, $\chi^2 = 4.365$, $p = 0.037$) as well as in the non-isolated ART group (1.47 *vs.* 6.43%, $\chi^2 = 11.988$, $p = 0.001$). These findings contradicted Sandra et al., who showed no significant increase of pathogenic CNVs in individuals born by assisted reproductive technology. However, the sample size in Monfort was very small, with only 34 samples (*Monfort et al., 2021*). Compared with the isolated ART group, the prevalence of pathogenic/likely pathogenic CNVs showed a nonsignificant increase in the non-isolated ART group (4.76 *vs.* 6.43%, $\chi^2 = 0.267$, $p = 0.606$). The finding is likely because ART caused microdeletion and microduplications easily. Jointly, the above data indicated CMA and karyotyping may provide further genetic data to enhance prenatal counselling and pregnancy management for all ART pregnant women, with or without other abnormalities.

Furthermore, VOUS have been classified as genomic anomalies (*Zhu et al., 2016*). However, VOUS might be nonmalignant CNVs and could not constitute anomalies. The present work revealed 2.38% of fetuses in the isolated ART group had VOUS, *versus* 3.57% in the non-isolated ART group. Previous large trials reported comparable rates of VOUS, ranging from 2% to 4% in fetuses showing abnormal ultrasound findings, (*Hillman et al., 2013*; *Shaffer et al., 2012*; *Wapner et al., 2012*) also in mothers with low-risk pregnancies (*Stern et al., 2021*). A comparable detection rate was obtained in this study.

In addition, we showed CMA was superior in detecting genetic abnormalities over karyotyping in cases with isolated ART with no other abnormalities. This conclusion is consistent with our hypothesis. In fetuses with isolated ART, CMA revealed pathogenic CNVs in 4.76% (4/84) of cases while karyotyping found no chromosomal abnormalities. Therefore, CMA provides substantial incremental genetic data, including four pathogenic CNVs and two VOUS, in comparison with karyotyping, and should be carried out in all ART pregnant women, with or without other abnormalities.

This study had many shortcomings. Firstly, its retrospective cross-sectional design may result in lower-quality data in comparison with prospective trials and cause population bias; however, all genetic findings were available in this study. Secondly, relatively few patients were examined; most ART-treated women refused invasive testing for CMA and karyotyping and would instead select NIPT, which does not cause miscarriage in contrast to chorion villus sample collection and amniocentesis. With this small sample size, there were very few instances of abnormal findings in pregnancies. Therefore, a prospective follow-up study is warranted. Finally, long-term postnatal follow-up of the born children with CNVs was not performed.

## CONCLUSIONS

The incidence rates of intra-chromosomal deletions or duplications (CNVs) and aneuploidies in fetuses conceived by assisted reproductive technologies have not been assessed so far. In this study, a 3.29% incremental pathogenic rate of CNVs was

found in the isolated ART group over control individuals consisting of naturally conceived fetuses without high risk. Additionally, CMA elevated the diagnostic yield of clinically relevant anomalies in fetuses conceived by assisted reproductive technologies compared with karyotyping. CMA and karyotyping are advocated in fetuses conceived by assisted reproductive technologies accompanied with or without additional ultrasound abnormalities. All pregnant women conceiving by assisted reproductive technologies should be aware of a significant CNV risk undetectable by biochemical analysis or current NIPT platforms, as well as the option for invasive prenatal assessment for CMA.

## ACKNOWLEDGEMENTS

We acknowledge our collaborators at the Medical Laboratory of Jining Medical University. We would like to thank Ran Jia who made suggestions for revisions and Xiu Zhang for the data management. In addition, we would like to thank Rui Sheng and all members of the prenatal diagnosis center for their support to the data collection.

### Funding
The authors received no funding for this work.

### Competing Interests
The authors declare there are no competing interests.

### Author Contributions
- Huan Guo conceived and designed the experiments, performed the experiments, analyzed the data, prepared figures and/or tables, authored or reviewed drafts of the article, and approved the final draft.
- Rui Sheng performed the experiments, prepared figures and/or tables, and approved the final draft.
- Xiu Zhang performed the experiments, prepared figures and/or tables, and approved the final draft.
- Xuemei Jin performed the experiments, prepared figures and/or tables, and approved the final draft.
- Wenjing Gu performed the experiments, authored or reviewed drafts of the article, and approved the final draft.
- Ting Liu performed the experiments, authored or reviewed drafts of the article, and approved the final draft.
- Haixin Dong performed the experiments, prepared figures and/or tables, and approved the final draft.
- Ran Jia conceived and designed the experiments, analyzed the data, authored or reviewed drafts of the article, and approved the final draft.

## Clinical Trial Ethics

The following information was supplied relating to ethical approvals (*i.e.*, approving body and any reference numbers):

The Ethics Committee of the Affiliated Hospital of Jining Medical University.

## Microarray Data Deposition

The following information was supplied regarding the deposition of microarray data:

File 3 and File 4.

## Data Availability

The raw measurements are available in the Supplementary Files.

## Supplemental Information

Supplemental information for this article can be found online at http://dx.doi.org/10.7717/peerj.14678#supplemental-information.

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
