# Peer review of "Prenatal diagnosis of fetuses conceived by assisted reproductive technology by karyotyping and chromosomal microarray analysis"

_PeerJ, doi:10.7717/peerj.14678_

## Round 0.1 · original submission · Minor Revisions

The authors are suggested to fix typo errors and include descriptive methodology and results sections.

As pointed out of by the reviewers, a clear rationale for the current study and elaborated discussion is highly recommended.

Reviewer 1 ·

Basic reporting

I request the authors to take another look at the manuscript for grammatical and typographical errors:
1) It would be helpful if the authors mentioned all the points presented from lines 77-84 as a paragraph instead of bullet points.
2) Typographical errors: Space is missing in line 74, in the term ‘onlyand’. In figure 1, VOUS is mentioned as VOUs. Please correct it.

Experimental design

No comment

Validity of the findings

No comment

Additional comments

For the benefit of the readers, here are my additional comments for further improvement of this manuscript.

1) The sample size is very small. With this sample size, there were very few instances of abnormal findings in pregnancies. I request the authors to comment on this aspect in more detail in the discussion section.

2) This is a retrospective cross-sectional study and I request the authors to talk more about this in the discussion section.

3) The authors have used the word ‘pathogenic’ throughout this manuscript. However, the abnormalities which they are studying in this manuscript are not caused by any pathogens. Hence, I request the authors to replace all instances of this term in the manuscript.

4) The term ‘local control individuals’ in the abstract is confusing. I request the authors to replace it with another term.

5) It is difficult to see the beginning of each sub-sections. For e.g., ‘Karyotype Analysis’, ‘CMA Analysis’, ‘Prenatal diagnostic karyotyping data’, etc. are just a few examples of subsection headings. I request the authors to format the headings of each subsection for better readability.

Reviewer 2 ·

Basic reporting

No comment

Experimental design

No comment

Validity of the findings

No comment

Additional comments

The following comments must be addressed.
1. More updates are needed for background information. It is necessary to include a clear hypothesis in the introduction
2. The methodology section needs to be rewritten
3. A clear description of sample collection is needed
4. Result descriptions were inadequate. It needs to be rewritten.
5. The discussion part should be rewritten in accordance with the hypothesis
6. The reference format should be consistent

Reviewer 3 ·

Basic reporting

No comment

Experimental design

No comment

Validity of the findings

No comment

Additional comments

This is an interesting manuscript in which the authors compared the karyotyping with CMA and shown that CMA elevated the diagnostic yield of clinically relevant anomalies in fetuses conceived by ART compared to karyotyping. It is very well written; I suggest describing the isolated and non-isolated ART at line 126 (even though it is mentioned in the figure) for the benefit of readers.

---

## Round 0.2 · accepted · Accept

The authors have addressed all the comments carefully.

Reviewer 1 ·

Basic reporting

The authors have addressed all the comments.

Experimental design

no comment

Validity of the findings

no comment

Additional comments

no comment